# Safety Evaluation of Heavy Metal Contamination and Pesticide Residues in Coix Seeds in Guizhou Province, China

**DOI:** 10.3390/foods11152286

**Published:** 2022-07-31

**Authors:** Jiaxing Yu, Xiangui Wang, Xiaolong Yao, Xiaomao Wu

**Affiliations:** 1Institute of Crop Protection, Guizhou University, Guiyang 550025, China; yujiaxing0813@163.com (J.Y.); ningmeng_564@163.com (X.W.); xiaolongyao1021@163.com (X.Y.); 2College of Agriculture, Guizhou University, Guiyang 550025, China; 3Guizhou Key Laboratory of Mountain Agricultural Diseases and Insect Pests, Guiyang 550025, China

**Keywords:** coix seed, heavy metal, pesticide residue, safety evaluation

## Abstract

The coix seed is a medicinal and edible plant with rich nutritional and medicinal values. With the expansion of the coix seed consumption market, the problem of coix seed safety has attracted attention worldwide. The aims of this work were to evaluate the contamination of mercury (Hg), lead (Pb), cadmium (Cd), arsenic (As), chromium (Cr) and 116 pesticides in coix seeds collected from 12 main producing regions of coix seeds in the Guizhou Province of China and to analyze the major contributors of heavy metal and pesticide contamination in coix seed. The results show that the average contents of Pb, Cd, As and Cr in the 123 coix seed samples were 0.0069, 0.0021, 0.0138 and 0.1107 mg/kg, respectively, while Hg was not detected in all coix seed samples. Among the five heavy metals detected, only the Cr contents of three samples were found to be higher than the contaminant limit of Chinese standard GB2762-2017 (CSGB). A total of 13 pesticides were detected in 29 samples from seven main production regions of coix seeds, accounting for 23.6% of all the samples. The detection rates of chlorpyrifos were the highest (8.13%), followed by fenpropathrin (4.06%), bifenthrin (2.43%) and phoxim (1.62%), while the detection rates of the remaining pesticides were below 1%. Moreover, the residual risk score of dichlorvos was the highest of all the pesticides detected. The pollution index and risk assessment of heavy metals and pesticide residues indicates that coix seeds were at safe levels for consumption. In the production process of coix seeds, the local government should control the soil in areas heavily polluted by heavy metals and strengthen the monitoring and guidance on the scientific and rational use of pesticides.

## 1. Introduction

Coix seed (*Coix lacryma-jobi* L. var. Adlay), as the seed kernel of the millet plant of the graminae family, is widely known for its rich nutritional and medicinal value [1]. It has been considered as the king of gramineae globally due to it being rich in carbohydrates, protein and essential amino acids for the human body and has various medicinal active ingredients, such as polysaccharides, alkaloids, and terpenoids [2]. Numerous studies have shown that long-term consumption of coix seeds balances the cholesterol content in human blood and prevents the probability of cardiovascular diseases such as myocardial infarction and atherosclerosis [3]. Coix seed extract can inhibit NFκB and protein kinase C signaling, which is a commonly available treatment for cancer in China [4]. In addition, coix seed, as a feed supplement, can improve growth performance and productivity of post-weaning pigs by reducing gut pH and modulating gut microbiota [5]. China is the main cultivation, production and consumption area of coix seed. The Guizhou Province of southwest China is the main production region of coix seed; the cultivation area reached 50,000 hm^2^ in 2021 [6]. The coix seed is exported to Japan, South Korea, the United States and other countries, with an annual export trade of about 2500 tons and about $12 million [7]. As the brand effect of coix seed continued to strengthen, the coix seed price rose to 15,000 RMB/ton in 2018–2019 [8].

Heavy metals and pesticides are at the top of the list of contamination toxins that harm coix seeds during production. As is known, the high concentrations of heavy metals and pesticides affect both soil and plants [9]. The accumulation of Pb in plants can cause different physiological and biochemical deficiencies, such as seed DNA damage, inhibition of seed germination, and reduction of chlorophyll [10,11]. Cd affects inhibition of mineral transportation, which causes a deficiency of minerals in plants [12]. Many studies have also shown that the accumulation of pesticides in plants can hinder plant growth and cause metabolic disorders. The exposure to insecticide chlorpyrifos inhibited nitrogen metabolism of mung beans [13]. The imidacloprid insecticide led to the decrease of many phytochemical substances in mustard plants [14].

Heavy metals and pesticides can be transmitted to the human body through the food chain and pose a serious threat to human health. Heavy metals can also produce mutagenic effects at very low concentrations. A number of human diseases, organ dysfunctions and malformations due to metal toxicity have been reported. For example, mercury poisoning may cause peripheral neuropathy, and lead poisoning may cause severe anemia and hemoglobulinuria [15]. The sources of heavy metals in soil are mainly influenced by natural and anthropogenic factors such as industrial production, fertilizer use, transportation and lithogenic input via weathering of parent materials and bedrocks [16]. Water-insoluble phosphorus fertilizers have been demonstrated to produce phosphate rocks, which play an important role in the immobilization of metals by precipitation as metal phosphates in the soil [17]. The major routes are respiration, skin contact and dietary intake for heavy metal entry into the human body; dietary intake is the main way due to high consumption. Moreover, the toxic pesticide residues can give rise to various chronic or acute diseases. For example, epidemiological studies suggest Parkinson’s disease is associated with pesticides [18]. Pesticides may disrupt the function of adipose tissue to promote obesity and metabolic diseases such as type 2 diabetes [19]. Therefore, as one of the main foodstuffs of human beings, the safety of coix seed products has attracted emerging attention.

In the present study, 123 coix seed samples were collected randomly from 12 main production regions of coix seeds in Guizhou Province. The analyses of lead (Pb), cadmium (Cd), chromium (Cr), arsenic (As), mercury (Hg) and 116 pesticide residues were carried out. Moreover, the safety of coix seeds was evaluated. This study aims to provide a scientific basis for coix seed risk management and decision making of local governments.

## 2. Materials and Methods

### 2.1. Sample Collection and Preparation

A total of 123 coix seed samples were collected randomly from 12 main production regions of coix seeds during the harvest period in the Guizhou Province of southwest China, and the distribution of sampling locations is shown in Figure 1. The sampling sites were recorded and denoted using latitude and longitude by selecting the WGS2000 coordinate system. Sampling was carried out at multiple points according to the plum point method, with 5 sub-points in each sampling plot. Each sample comprised 3 kg of coix seed, and the collected samples were mixed into a mixed sample. After being air-dried, the samples were dehusked, ground into fine flour and sieved through a 0.43 mm mesh (40 Mesh).

### 2.2. Reagents and Instruments

Reagents: Single-element standard solutions of As, Pb, Cd and Cr at the concentration of 1000 µg/mL were purchased from Guobiao (Beijing, China) Testing & Certification Co., Ltd. Hg standard solution (100 µg/mL) was supplied by National Institute of Metrology, Beijing, China. A total of 116 certified pesticide standard solutions were obtained from Environmental Quality Supervision, Inspection and Testing Center of Ministry of Agriculture (Tianjin, China). Analytical grade reagents such as ethyl acetate, nitric acid, acetonitrile, n-hexane, toluene and sodium chloride (NaCl) were purchased from Merck.

Instruments: The following were used in the analysis: Aglient 7800 inductively coupled plasma mass spectrometry (ICP-MS), AFS-230E atomic fluorescence spectrometry (AFS), MAR6 microwave digester, Waters XEVO-TQXS ultra performance liquid chromatography-tandem mass spectrometry (UPLC-MS/MS), TSQ 8000 gas chromatography-tandem mass spectrometry (GC-MS/MS), FW100 disintegrator, Milli-Q Ultra-pure Water System.

### 2.3. Analyses of Heavy Metals

#### 2.3.1. Sample Digestion and Determination of Element Concentrations

The sample preparation of As, Cd, Cr and Pb followed Chinese standard GB5009.268-2016 with minor modifications. The sample pretreatment of Hg was carried out in accordance with the Chinese standard GB 5009.17-2014 with minor modifications. Firstly, 0.5 g of sample was levigated and sieved (accurate to 0.001 g), and 6 mL nitric acid was put into digestion tank for 1 h. Secondly, 2 mL hydrogen peroxide was added into the digestion tank, and then the digestion tank was put into a microwave digester for digestion. Digestion was performed under the following program: ramped to 80 °C (held for 2 min), ramped to 120 °C (held for 2 min), ramped to 150 °C (held for 2 min), ramped to 180 °C (held for 2 min) and ramped to 200 °C (held for 30 min). After digestion, the tank was washed with water, and the wash solution was put through a 0.22 μm filter. Finally, the samples were analyzed for As, Cd, Cr and Pb by ICP-MS and for Hg by AFS.

The detection conditions of AFS for Hg were as follows: lamp current, 45 mA; negative high voltage, 235 V; reading time, 14 s; carrier gas flow rate, 300 mL/min; shielding air flow rate, 800 mL/min. The detection conditions of ICP-MS for As, Pb, Cd and Cr were as follows: RF power 1550 W; plasma gas flow rate, 14 L/min; carrier gas flow rate, 0.97 L/min; auxiliary air flow rate, 0.80 L/min; helium flow rate, 4.5 L/min; atomizing chamber temperature, 2 °C; sample uptake flow rate, 0.3 r/s; sampling depth, 8.00 mm.

#### 2.3.2. Assessment Methods

Pollution Index

Evaluation of heavy metal pollution degree was carried out in accordance with Chinese standard GB2762-2017 (CSGB) and Chinese pharmacopoeia (CP). The national standard does not specify the limit of coix seed but uses the limit standard of cereals and coarse cereals as the evaluation basis.

Individual factor index (Pi) of each heavy metal was calculated according to the following equation:(1)Pi=Ci/Si
where Pi represents the single factor pollution index of element i, Ci represents the measured data of element i in coix seed, and Si represents the limited standard value of element i in coix seed. Pi < 1 means that the coix seed is safe, whereas Pi > 1 means that the coix seed is contaminated by heavy metals.

The pollution factor analysis can indicate the main heavy metal pollution factors in coix seed. The calculation equation is as follows:(2)L=Pi/∑Pi
where L is the pollutant load ratio, and Pi is the single factor pollution index of heavy metal i.

The Nemerow integrated pollution index (PN) is used to comprehensively evaluate heavy metal pollution in coix seed. The calculation equation is as follows:(3)$$PN=\sqrt{\frac{p2\sum_{}{}_{i\max}+p2\sum {}_{iave}}{2}}$$
where PN represents the comprehensive pollution of multiple heavy metals, Pimax is the maximum single factor pollution index of heavy metal i, and Piave is the mean of the single factor pollution index of heavy metal i. The PN was set up in different classes: Ⅰ (PN ≤ 0.7), Ⅱ (0.7 < PN ≤ 1), Ⅲ (1 < PN ≤ 2), Ⅳ (2 < PN ≤ 3), and Ⅴ (PN > 3), which are clean level, warning line level, light pollution level, moderate pollution level, and heavy pollution level, respectively [20].

2.Enrichment Characteristic

The enrichment coefficient of heavy metals reflects the enrichment capacity of crops for different heavy metals, which can prevent heavy metal pollution, and the calculation equation is as follows:(4)BCF=Cp/Cs
where BCF is the bioconcentration factors, Cp is the concentration of heavy metals in crops, and Cs refers to the concentration of the heavy metal in soil [21].

3.Human Health Risk

The health risk of heavy metals (As, Cd, Cr, Pb, Hg) was evaluated for hazard quotient (HQi) and hazard index (HI) using the United States Environmental Agency (US-EPA) standard, as follows [22]:(5)$$HQi=\frac{Ci\times IR\times ED\times EF}{BW\times RfD\times AT}$$
where Ci is the concentration of heavy metal i (mg/kg); IR is the ingestion rate of coix seed (0.0311 kg/day); ED is the total exposure duration (70 years); EF is the exposure frequency (282 days/year); BW is the body weight for adults (estimated to be approximately 60 kg); RfD, set by the US-EPA, is the reference dose; RfD for As, Cd, Cr, Pb and Hg is 3 × 10^−4^, 1 × 10^−3^, 3 × 10^−3^, 3.5 × 10^−3^ and 3 × 10^−4^ mg/kg/day, respectively; AT is the average exposure time (70 days × 282 days/year).
(6)$$HI=\sum_{i=1}^{n}HQi$$
where n is the total of heavy metals evaluated for a health risk assessment; HI ≤ 1 indicates no adverse effects; HI > 1 indicates noncarcinogenic adverse effects; HI ≥ 10 indicates chronic toxic effects [23].

### 2.4. Analyses of Residual Agricultural Chemicals

#### 2.4.1. Determination of Concentrations of 116 Pesticides

The QuEChERS method was employed in the sample preparation procedure for determination of pesticide residues. Samples (2.5 ± 0.01 g) were deposited into a 50 mL centrifuge tube; 10 mL of ultrapure water followed by 25 mL of acetonitrile was added, and the mixture was blended by vortex oscillator for 5 min. Thereafter, 1 g of NaCl was added, and the tubes were blended immediately by vortex oscillator for 2 min and then centrifuged at 2000 rpm for 5 min. A volume of 2 mL supernatant was collected into 10 mL centrifuge tubes and then diluted with 1.0 mL of methanol and water (1 + 1, *v/v*). The clean extract was filtered through a 0.22 µm nylon filter and finally analyzed by UPLC-MS/MS. For GC-MS/MS analysis, solid phase extraction (SPE) was used to achieve extraction and purification of coix seed samples. A quantity of 2.5 ± 0.01 g of coix seed sample was put in a 50 mL centrifugation tube with 10 mL of ultrapure water. After being blended by vortex oscillator, the tube was allowed to stand for 30 min and 25 mL of acetonitrile was added. Then, it was homogenized at 15,000 rpm for 2 min in a high-speed homogenizer. Thereafter, 5~6 g of NaCl was added, and the tube was shaken for 1 min and then centrifuged at 4200 rpm for 5 min. Finally, 5.00 mL supernatant was purified by SPE containing composite amino column. Then, 5 mL acetonitrile and toluene (3 + 1, *v/v*) was added to activate the composite amino column, and the supernatant was loaded onto the column in two portions and the liquid was collected in a 250 mL pear-shaped bottle. The composite amino column was washed five times with 25 mL of acetonitrile and toluene (3 + 1, *v/v*). All the lavage liquids collected were concentrated to near dryness on a rotary evaporator at 40 °C and diluted to 1 mL with ethyl acetate, then filtered for GC-MS/MS analysis.

Fifty-six compounds were analyzed by UPLC-MS/MS (listed in Table A1, Appendix B). The column temperature was maintained at 40 °C, and the injection volume was 2.0 μL. The flow rate was set to 0.3 mL/min. The mobile phase gradient consisted of 0.1% formic acid in water (A) and methanol (B). The elution gradient started at 10% B and increased linearly to 40% B over 1 min, then increased to 60% B for 3 min and to 80% B for 4 min, rose to 97% B for 2 min, reached 100% B for 1 min, and then decreased to 10% B for 0.1 min and was kept at 100% B for 2.9 min. The ionization parameters were set to: capillary voltage 3.0 kV, desolvation temperature 500 °C, desolvation gas flow rate 800 L/h, source temperature 150 °C and cone gas flow rate 150 L/h. The analysis was performed in multiple-reaction-monitoring mode, and argon was used as the collision gas.

Sixty compounds were analyzed by GC-MS/MS (listed in Table A2, Appendix B). The column oven temperature program was 50 °C (held for 2 min), ramped to 150 °C at 50 °C/min, ramped to 200 °C at 5 °C/min and finally ramped to 300 °C at 15 °C/min (held for 3 min). The carrier gas was helium (99.999% purity), with a constant flow rate of 1.4 mL/min maintained, and the injection volume was 2.0 μL. MS parameters such as transfer line temperature and ion source temperature were 300 °C. The energy ionization was 70 eV, and the MRM scan mode was selected.

#### 2.4.2. Assessment Methods

1.Chronic Dietary Exposure Assessment

The chronic dietary intake risk of each pesticide was determined using the national estimated daily intake (NEDI) and acceptable daily intake (ADI) [24,25]. The daily dietary exposure of a pesticide was calculated based on the following equation:(7)$$NEDI=\frac{STMR\times F_{}}{bw}$$
where NEDI is the national estimated daily intake (mg/kg·d); STMR is the median residue, taking the average residual value (mg/kg); F is the average daily consumption of coix seed (kg/d)—31.1 g of coix seed per adult per day according to the survey; bw is average weight for adults (estimated to be approximately 60 kg).
(8)$$\%ADI=\frac{NEDI}{ADI}\times 100\%$$
where %ADI is the chronic ingestion risk; ADI is the acceptable daily intake of a pesticide, mg/kg·d, with the ADI value taken from GB2763-2021 (National Health Agency of China 2021). If %ADI ≤ 100% means the chronic dietary intake risk of the pesticide is in the acceptable range, %ADI > 100% means the chronic intake risk is unacceptable. The smaller the %ADI is, the smaller the chronic intake risk is.

2.Acute Exposure Assessment


(9)
$$NESTI=\frac{LP\times HR}{bw}$$



(10)
$$\%ARfD=\frac{NESTI}{ARfD}$$


The acute risk was calculated using Equations (9) and (10). NESTI (mg/kg d) represents the estimated short-term intake; LP is the large portion of coix seed consumption in Chinese population (according to the survey feedback, 60.4 g is taken); HR (mg/kg) is the highest amount of pesticide residues in coix seed, and the 99.5 percentile residue value was taken in this study; ARfD (mg/kg·d) is the acute reference dose. If the %ARfD value is lower than 100%, the exposure risk is acceptable. The higher the value is, the greater the risk is. When the value is higher than 100%, it indicates an unacceptably high risk to consumers [26,27].

3.Pesticide Residue Risk Ranking

Based on the risk ranking matrix of veterinary drug residues of the UK Veterinary Drug Residue Committee, the pesticide risks were ranked by six indicators: pesticide hazard, toxic effect, dietary ratio, frequency of pesticide use, presence of highly exposed populations and residue levels [28]. The original assignment criteria were used in this study, as shown in Appendix A. Frequency of pesticide use (FOD) is calculated according to Equation (11). The residual risk score (S) of each pesticide in the sample was calculated using Equation (12). The higher the residue level of a pesticide in the sample is, the greater the risk score, and the greater the threat to human health.
(11)FOD=T/P×100
(12)S=(A+B)×(C+D+E+F)
where P is the time from bud to maturity of coix seed, expressed by day; T is the number of times the pesticide is used during the growth of coix seed; A, B, C, D, E and F indicate the scores of toxicity, toxic effect, dietary proportion, frequency of pesticide use, highly exposed population, and residue level, respectively.

### 2.5. Validation of Analytical Methods

The quality assurance of analytical methods for heavy metals was accomplished by analyzing reference materials, which were summarized in Table 1. The relative error between the determination results of standard samples and the reference value was within the guideline range, and the determination results are satisfactory according to the Chinese standard GB/T 27404-2008. All the standard curves showed good linearity, and the regression coefficient (R^2^) was between 0.9991 and 0.9998.

The quality assurance of analytical methods for pesticide residues was accomplished by analyzing reference materials and adding three concentrations (0.01, 0.1, 1.0 mg/kg) of reference materials to the blank samples, which did not contain detectable concentrations of pesticides. The validation results of pesticides analyzed are detailed in Table A3 and Table A4 in Appendix C. The linear correlation coefficients of 56 pesticides by UPLC-MS/MS ranged from 0.9887 to 0.9999, the method limits of quantification (*LOQ*s) were all 0.01 mg/kg, the recoveries of three concentrations added at different levels were all between 81.1%~117.2%, and the relative standard deviations (*RSD*s) were between 0.4% and 13.9%. The linear correlation coefficients of the 60 pesticides by GC-MS/MS ranged from 0.9902 to 0.9997, the average recoveries of the three concentrations at different levels were between 74.6% and 117.6%, *RSD*s ranged from 0.9% to 19.3%, and *LOQ*s were also all 0.01 mg/kg. The present results show that the recoveries and precision of the pesticide residue analysis were satisfactory (Chinese standard NYT 788-2018).

### 2.6. Statistical Analysis

The data obtained were statistically analyzed by ANOVA using the DPS software package (version 7.05, Hangzhou, China). The differences between the means of data were determined by Duncan’s multiple range test at *p* < 0.05 [29]. Halves of the limit detection values were used for statistical analysis in all undetected results, as recommended by the World Health Organization (WHO) (Geneva, Switzerland) [30].

## 3. Results and Discussion

### 3.1. Heavy Metals in Coix Seed

#### 3.1.1. Concentrations of Heavy Metals

The analysis of 123 samples collected from the 12 main production regions of coix seeds in Guizhou Province show there were differences in the contents of hazardous elements in the coix seeds. The differences in the contents of five heavy metals were not the same in the different sampling regions of Guizhou Province (Table 2). The element Hg was not detected in coix seeds of the 12 main production regions. The lowest and highest average contents of Cd were detected in Guanling (0.0005 mg/kg) and Yilong (0.0201 mg/kg), respectively. The lowest average contents of Pb were in samples from Panzhou, Guanling and Puan, at 0.0050 mg/kg, while the highest average contents were in samples from Zhenan, at 0.0520 mg/kg. The lowest and highest average contents of As were found in samples from Guanling (0.0028 mg/kg) and Yilong (0.1463 mg/kg), respectively. The lowest and highest average contents of Cr were from samples from Xingyi (0.0386 mg/kg) and Puding (0.3560 mg/kg), respectively.

The statistical results of each heavy metal concentration are displayed in Table 3. The mean content of Cr in coix seeds was higher than that of the other heavy metals, followed by As. The content of Cd, As, Cr and Pb in coix seeds was varied in the 12 sampling regions, with variation coefficients ranging from 89.94% to 291.46%. Among the 123 coix seed samples, the Hg element was not detected, while Cr content in three samples collected from Xingren was higher than the contaminant limits of CSGB. The high background value of Cr in soil might be the main reason for its content exceeding the contaminant limits, which is consistent with previous research conclusions [31]. The concentration of heavy metals in coix seeds might be influenced by different irrigation sources. The crops irrigated using wastewater were characterized by high concentrations of heavy metals [32,33]. It has been reported that the altitude influences the distribution of heavy metals in the soil, which therefore affects the absorption of heavy metals by crops [34,35]. In addition, it was possible that the pollution of heavy metals in coix seeds was related to rainfall, atmospheric deposition and stomata on leaves [36].

#### 3.1.2. Heavy Metal Pollution Assessment

Since only four heavy metals, Cd, Pb, As and Hg, were specified in CP, the contamination level of Cr was only evaluated using CSGB in the process of contamination degree evaluation. As shown in Figure 2, regardless of the contamination limits specified in CP or CSGB as the reference standard, the Pi values of each heavy metal were less than 1, and the coix seed was in a pollution-free state.

As exhibited in Table 4, L values in the coix seed decreased in the order of Pb > As > Cd > Hg, using CP as the evaluation standard, while the order of L values was Cr > Pb > Hg > As > Cd, using CSGB as the evaluation standard. Pb and Cr were the main heavy metal pollution factors in coix seeds, as compared with other heavy metals. It can be seen from the data in Table 4 that the average pollution index (PI) of five heavy metals was <0.1, which indicates that coix seed was not polluted.

In addition, the Nemerow integrated pollution index method was used to evaluate the pollution of Cr, Pb, Hg, As and Cd in coix seeds using CP or CSGB as the evaluation standard; the PN value of five heavy metals mentioned above was <0.7, which shows that the coix seed was clean and safe to human health (Table 4).

#### 3.1.3. Evaluation of Enrichment Characteristics of Heavy Metals

As indicated in Figure 3, the BCF order of the five heavy metals was Hg < Pb < As < Cr < Cd. The adsorption capacity of the heavy metal Cd in coix seeds was relatively strong, while that of Hg was the weakest, followed by Pb as the second-weakest. There was no Hg element detected in 123 coix seed samples, which implied that weaker enrichment capacity was an important factor affecting the adsorption of the Hg element. The content of heavy metals in soil was positively correlated with heavy metal pollution in coix seeds, except for Cd (Figure 3). High background values of heavy metals in local soils were easily enriched in crops [37].

#### 3.1.4. Human Health Risk Assessment

The risk level of heavy metals to human health in the 12 main production regions of coix seed was investigated. As shown in Table 5, the risk of a single heavy metal in 12 regions was relatively small, with HQ and HI values less than 1, which means that five heavy metals have no significant safety risks to consumers.

### 3.2. Pesticide Residues in Coix Seeds

#### 3.2.1. Detection of 116 Pesticides

Pesticides were detected in 29 samples, accounting for 23.6% of 123 analyzed samples from 12 main production areas of coix seeds. There were 13 pesticides detected in seven coix seed production areas, five pesticides of which were of moderate toxicity, namely chlorpyrifos, profenofos, fenpropathrin, dichlorvos and bifenthrin. The remaining eight pesticides were of low toxicity, namely trichlorfon, imidacloprid, phoxim, pyridaben, propiconazole, difenoconazole, azoxystrobin and tebufenozide. As shown in Table 6, the detection rate (8.13%) of chlorpyrifos was the highest among all the detected pesticides, followed by fenpropathrin, bifenthrin and phoxim at 4.06%, 2.43% and 1.62%, respectively. The detection rate of other pesticides was below 1%.

There were four coix seed samples whose pesticide residue concentration exceeded the maximum residue limits (MRLs) of GB2763-2021: phoxim, chlorpyrifos, trichlorfon and difenoconazole, with a concentration of 0.051, 1.47, 0.1255 and 0.41 mg/kg, respectively. The pesticide residues of the remaining 119 coix seed samples were below MRLs. The pesticides mentioned above are widely used in the control of agricultural pest insects and diseases. Chlorpyrifos is known as one of the most widely used organophosphate insecticides; it can effectively control more than one hundred pest insects in a variety of crops such as grains, apples and vegetables [38]. *Coix lacryma-jobi* L, a high straw crop, has the strong ability of tiller branching and needs plant growth regulators to achieve thick roots and strong plants in growing. Toxic substances in the soil are easily transported to the crop through the roots during irrigation. In addition, the main reason for pesticide residues in coix seeds might be the irrational use of pesticides. On one hand, in order to protect their crops from pest insects and diseases, farmers might frequently use pesticides during the growth season of coix seeds. On the other hand, the application doses of pesticides were significantly higher than their recommended doses during the growth season of coix seeds.

#### 3.2.2. Risk Assessment

The dietary intake risk of each pesticide is shown in Table 7. The chronic dietary intake risk (% ADI) of 13 pesticides was very low, between 0.003% and 3.253%, far less than 100%; the average was only 0.607%. This means that the risk of chronic dietary intake of coix seed pesticide residues in Guizhou province is very low, and consumers need not worry about the threat to human health.

The ARfD information of 10 pesticides is presented in Table 7 (no ARfD information was available for azoxystrobin, phoxim and pyridaben) [39]. Based on the ARfD values (Table 7), the acute dietary intake risk for the remaining 10 pesticides ranged from 0.002% to 1.415%, with a mean value of 0.3216%, which was much less than 100%. Among them, %ARfD values of chlorpyrifos and bifenthrin were higher, at 1.415% and 0.748%, respectively. On the basis of the safety threshold data, the acute dietary risk of these 10 pesticides does not pose a threat to human health, and thus it is safe for human beings to consume coix seeds.

As exhibited in Figure 4, the risk ranking of the 13 pesticides in coix seeds can be divided into three categories: the first category is the high-risk pesticide A, which includes difenoconazole, bifenthrin, dichlorvos, chlorpyrifos and trichlorfon, with a risk score greater than 20; the second category is the medium-risk pesticide B, which has a risk score greater than 15, which includes phoxim; the third category is the low-risk pesticide C, which has a risk score greater than or equal to 10, which includes imidacloprid, profenofos, pyridaben, fenpropathrin, propiconazole, tebufenozide and azoxystrobin.

## 4. Conclusions

The contamination of heavy metals (Hg, Pb, Cd, As and Cr) and 116 pesticides in 123 coix seed samples collected from 12 main production regions of coix seeds in Guizhou Province of southwest China was detected, and their safety was also assessed. The average levels of Pb, Cd, As and Cr in coix seeds were 0.0069, 0.0021, 0.0138 and 0.1107 mg/kg, respectively, while Hg was not detected in all coix seed samples. The Cr contents of three samples exceeded the safety standard of CSGB. On the contrary, the result of pollution index and human risk assessment indicates the coix seed was still at a clean level, and the contents of heavy metals detected do not present a significant health risk to consumers. The detection rates of pesticides in 123 coix seeds accounted for 23.6% of all samples. There were 13 pesticides detected, of which chlorpyrifos had the highest detection rate of 8.13%, with the residual level of 0.011~1.47 mg/kg. Moreover, the residual risk score of dichlorvos was the highest of all the pesticides detected. However, the assessment results of chronic and acute dietary exposure indicate the pesticide residues of coix seeds do not pose a threat to human health and that it is safe for human beings to consume coix seeds.

Overall, this work is the first systematic analysis and safety evaluation of five heavy metals and 116 pesticides in coix seeds from main production regions in Guizhou Province; it is of great significance for local authorities as a reference for toxic substance control in coix seeds. Although there are indications that the risk of coix seed may be low, these risk assessments cannot be ignored, and thus additional monitoring of exposure to heavy metals and chemical pesticides should be done. Sampling of coix seed at different growth periods to explore the distribution of contaminants in various parts of the coix seed can help develop contaminant control measures for the future of human consumption.

In the future production of coix seeds, agricultural departments should conduct environmental evaluation of coix seed planting sites before planting and remediate the soil in areas heavily polluted by heavy metals. Organic fertilizers and chemical fertilizers with low heavy metal content should be promoted. In addition, agricultural departments should strengthen the monitoring of and guidance on the scientific and rational use of pesticides. The comprehensive control techniques of the coix seed pests should be adopted to reduce the use of pesticides.

## Figures and Tables

**Figure 1 foods-11-02286-f001:**
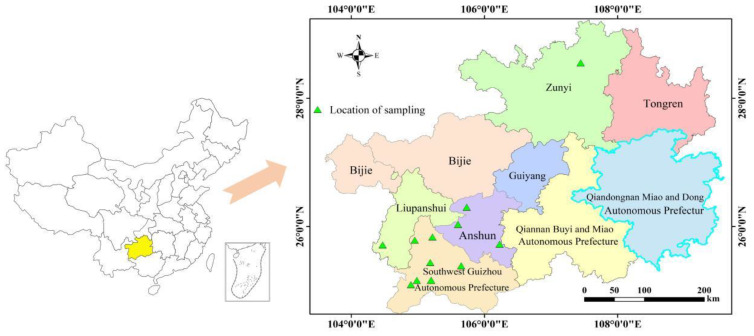
The distribution of sampling sites.

**Figure 2 foods-11-02286-f002:**
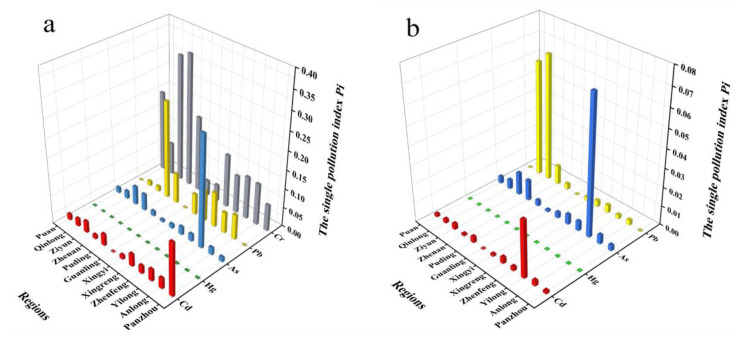
The single pollution index was conducted with reference to CSGB (**a**) and CP (**b**).

**Figure 3 foods-11-02286-f003:**
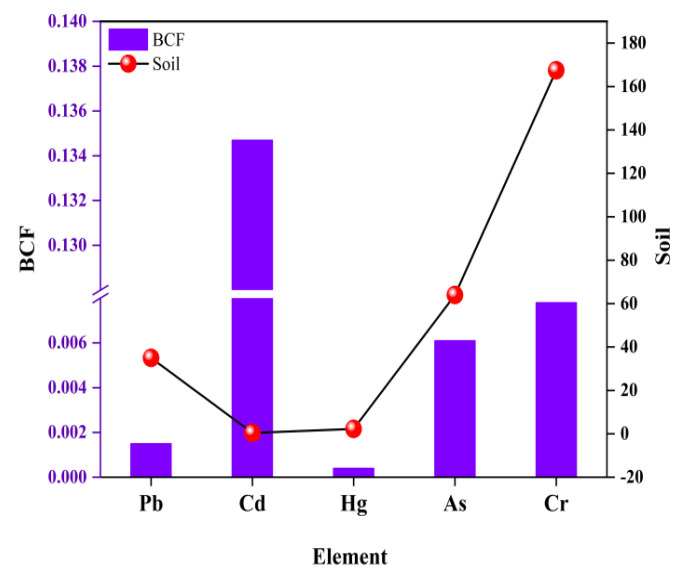
The enrichment evaluation of heavy metals in coix seeds.

**Figure 4 foods-11-02286-f004:**
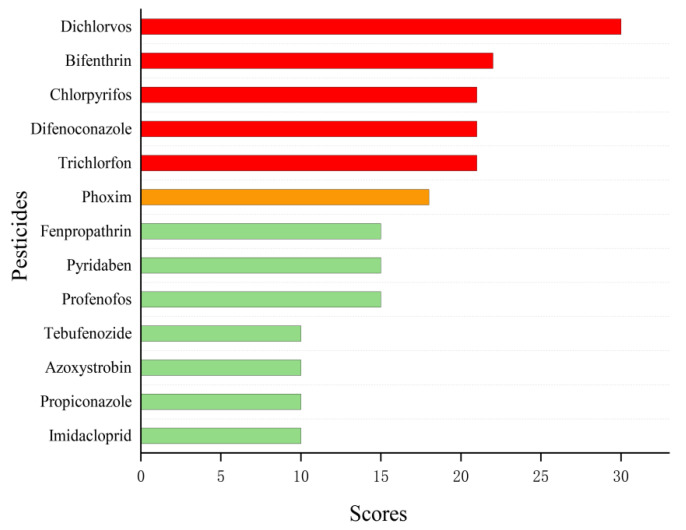
The risk ranking of residues of 13 pesticides in coix seeds.

**Table 1 foods-11-02286-t001:** The validation parameters of the analytical method of heavy metals.

Elements	Detection Limit (μg/L)	R^2^	Reference Value (mg/kg)	Detected Value ^a^ (mg/kg)	Relative Error (%)	*RSD* (%)
As	0.001	0.9995	0.0280 ± 0.0060	0.0320	−17.9	3.12
Cd	0.001	0.9996	0.0041 ± 0.0016	0.0050	−25.0	3.92
Cr	0.03	0.9998	0.1100 ± 0.0000	0.1070	−8.2	5.43
Hg	0.002	0.9991	0.0016 ± 0.0000	0.0016	0.0	6.25
Pb	0.01	0.9997	0.0700 ± 0.0200	0.0740	−10.0	4.17

Key: ^a^ mean value from triplicate determinations.

**Table 2 foods-11-02286-t002:** The mean contents of five heavy metals in coix seeds from 12 cities in Guizhou Province.

Region	Cd (mg/kg)	Pb (mg/kg)	As (mg/kg)	Cr (mg/kg)
Panzhou	0.0014 ^b^ ± 0.0004	0.0050 ^c^ ± 0.0000	0.0059 ^b^ ± 0.0016	0.0694 ^a^ ± 0.0694
Anlong	0.0018 ^b^ ± 0.0019	0.0066 ^bc^ ± 0.0036	0.0094 ^b^ ± 0.0063	0.0889 ^a^ ± 0.0694
Yilong	0.0201 ^a^ ± 0.0324	0.0090 ^bc^ ± 0.0035	0.1463 ^a^ ± 0.2103	0.1131 ^a^ ± 0.0940
Zhenfeng	0.0011 ^b^ ± 0.0007	0.0095 ^bc^ ± 0.0071	0.0094 ^b^ ± 0.0047	0.0709 ^a^ ± 0.0587
Xingreng	0.0017 ^b^ ± 0.0041	0.0062 ^bc^ ± 0.0048	0.0107 ^b^ ± 0.0126	0.1083 ^a^ ± 0.2101
Xingyi	0.0011 ^b^ ± 0.0004	0.0067 ^bc^ ± 0.0029	0.0053 ^b^ ± 0.0021	0.0386 ^a^ ± 0.0191
Guanling	0.0005 ^b^ ± 0.0000	0.0050 ^c^ ± 0.0000	0.0028 ^b^ ± 0.0000	0.0435 ^a^ ± 0.0000
Puding	0.0025 ^b^ ± 0.0000	0.0160 ^b^ ± 0.0000	0.0100 ^b^ ± 0.0000	0.3560 ^a^ ± 0.0000
Zhenan	0.0013 ^b^ ± 0.0000	0.0520 ^a^ ± 0.0000	0.0219 ^b^ ± 0.0000	0.3520 ^a^ ± 0.0000
Ziyun	0.0031 ^b^ ± 0.0012	0.0078 ^bc^ ± 0.0048	0.0247 ^b^ ± 0.0256	0.3373 ^a^ ± 0.0647
Qinlong	0.0022 ^b^ ± 0.0006	0.0068 ^bc^ ± 0.0036	0.0101 ^b^ ± 0.0049	0.0687 ^a^ ± 0.0496
Puan	0.0015 ^b^ ± 0.0006	0.0050 ^c^ ± 0.0000	0.0073 ^b^ ± 0.0018	0.2158 ^a^ ± 0.1870

Key: Mean concentrations of each heavy metal in coix seed collected from different coix seed producing regions with different letters are significantly different (*p* < 0.05).

**Table 3 foods-11-02286-t003:** The statistics of heavy metal concentrations in coix seeds.

Element	Min ^a^ (mg/kg)	Max (mg/kg)	Mean	CV (%)	Chinese Standard (CSGB)		Chinese Pharmacopoeia (CP)	
					Limit (mg/kg)	EN	Limit (mg/kg)	EN
Cd	0.0005	0.0575	0.0021 ^b^ ± 0.0062	291.46	0.1	0	1.0	0
Hg	0.0010	0.0010	0.0010 ^b^ ± 0.0000	0.00	0.02	0	0.2	0
As	0.0017	0.3880	0.0138 ^b^ ± 0.0361	262.19	0.5	0	2.0	0
Cr	0.0150	1.3000	0.1107 ^a^ ± 0.1879	169.71	1.0	3	-	0
Pb	0.0050	0.0520	0.0069 ^b^ ± 0.0062	89.94	0.2	0	5.0	0

Key: ^a^ Target analyses with concentrations lower than detection limit were treated as one-half of detection limit when calculating the mean and minimum values. Mean concentrations of different heavy metals in coix seed with different letters are significantly different (*p* < 0.05). CV, coefficient of variation; EN, exceeding the standard number.

**Table 4 foods-11-02286-t004:** The evaluation of heavy metal pollution degree.

Element	CSGB				CP			
	Max	PI	L	PN	Max	PI	L	PN
Cd	0.2008	0.0667	0.0768 ^d^	0.1864	0.0201	0.0053	0.0812 ^c^	0.0275
Hg	0.0500		0.0000 ^e^		0.005		0.0000 ^d^	
As	0.2926		0.0845 ^c^		0.0732		0.0845 ^b^	
Cr	0.3520		0.0912 ^b^		-		-	
Pb	0.2600		0.0982 ^a^		0.0104		0.0982 ^a^	

Key: Max is the maximum Pi value of heavy metals; L values of different heavy metals in coix seed with different letters are significantly different (*p* < 0.05); PI, average pollution index.

**Table 5 foods-11-02286-t005:** The risk evaluation of heavy metals in coix seeds.

Region	HQ					HI
	Cd	Hg	As	Pb	Cr	
Panzhou	0.001 ^c^	0.002 ^a^	0.010 ^fg^	0.001 ^c^	0.012 ^h^	0.026 ^j^
Anlong	0.001 ^c^	0.002 ^a^	0.016 ^de^	0.001 ^c^	0.015 ^g^	0.035 ^g^
Yilong	0.01 ^a^	0.002 ^a^	0.252 ^a^	0.001 ^c^	0.020 ^e^	0.285 ^a^
Zhenfeng	0.001 ^c^	0.002 ^a^	0.016 ^de^	0.001 ^c^	0.012 ^h^	0.032 ^i^
Xingren	0.001 ^c^	0.002 ^a^	0.018 ^d^	0.001 ^c^	0.019 ^f^	0.041 ^f^
Xinyi	0.001 ^c^	0.002 ^a^	0.011 ^efg^	0.001 ^c^	0.009 ^i^	0.024 ^k^
Guanling	0.0003 ^ce^	0.002 ^a^	0.007 ^g^	0.001 ^c^	0.009 ^i^	0.019 ^l^
Puding	0.001 ^c^	0.002 ^a^	0.017 ^d^	0.002 ^b^	0.062 ^a^	0.084 ^d^
Zhengan	0.001 ^c^	0.002 ^a^	0.038 ^c^	0.007 ^a^	0.06 ^b^	0.108 ^b^
Ziyun	0.002 ^b^	0.002 ^a^	0.043 ^b^	0.001 ^c^	0.058 ^c^	0.106 ^c^
Qinglong	0.001 ^c^	0.002 ^a^	0.017 ^d^	0.001 ^c^	0.012 ^h^	0.033 ^h^
Puan	0.001 ^cd^	0.002 ^a^	0.013 ^def^	0.001 ^c^	0.037 ^d^	0.054 ^e^

Key: HQ of each heavy metal in coix seed collected from different coix seed producing regions with different letters are significantly different (*p* < 0.05). HI of heavy metals collected from different coix seed producing regions with different letters are significantly different (*p* < 0.05).

**Table 6 foods-11-02286-t006:** Pesticide residue levels in coix seed.

Pesticide	Toxicity	Number of Residual Samples	Detection Rate (%)	Residual Level (mg/kg)	Risk Score
Trichlorfon	low	1	0.81	0.1255	21
Imidacloprid	low	1	0.81	0.028	10
Phoxim	low	2	1.62	0.0161~0.051	18
Tebufenozide	low	1	0.81	0.0355	10
Chlorpyrifos	moderate	10	8.13	0.011~1.47	21
Profenofos	moderate	1	0.81	0.019	15
Pyridaben	low	1	0.81	0.016	15
Fenpropathrin	moderate	5	4.06	0.014~0.22	15
Dichlorvos	moderate	1	0.81	0.037	30
Propiconazole	low	1	0.81	0.0254	10
Bifenthrin	moderate	3	2.43	0.012~0.15	22
Difenoconazole	low	1	0.81	0.41	21
Azoxystrobin	low	1	0.81	0.0656	10

**Table 7 foods-11-02286-t007:** The risk assessment of pesticide residues.

Pesticide	Chronic Risk Assessment			Acute Risk Assessment		
	ADI (mg/kg)	%ADI (%)	Max (mg/kg)	ARfD (mg/kg)	%ARfD (%)	Safety Margin (mg/kg)
Phoxim	0.004	0.435	0.051	-	-	-
Trichlorfon	0.002	3.253	0.1255	0.1	0.126	99
Imidacloprid	0.06	0.024	0.028	0.4	0.007	397
Chlorpyrifos	0.01	0.865	1.4700	0.1	1.415	99
Profenofos	0.03	0.033	0.0190	1.0	0.002	993
Pyridaben	0.01	0.083	0.0160	-	-	-
Fenpropathrin	0.03	0.162	0.2200	0.03	0.730	30
Dichlorvos	0.004	0.479	0.0370	0.1	0.037	99
Propiconazole	0.07	0.019	0.0254	0.3	0.009	298
Bifenthrin	0.01	0.314	0.1500	0.02	0.748	20
Azoxystrobin	0.2	0.003	0.0656	-	-	-
Tebufenozide	0.02	0.092	0.0355	0.9	0.004	894
Difenoconazole	0.01	2.125	0.4100	0.3	0.138	298

## Data Availability

Data is contained within the article or Appendix A.

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
