# Peer review of "Safety Evaluation of Heavy Metal Contamination and Pesticide Residues in Coix Seeds in Guizhou Province, China"

_foods, 2022, doi:10.3390/foods11152286_

Round 1
Reviewer 1 Report
The presented study entitled “Safety evaluation of heavy metals contamination and pesticide residues in coix seeds in Guizhou Province, China”, investigates different samples of this graminae analysing heavy metal and pesticide contents. The novelty of the study is based on the regional screening information and the evaluation of different safety indexes. In general, the study is well structured and executed, especially the analytical issues.
The following suggestions should be considered by the authors:
- Line 25: “at clean level” this term should be change by a more precise term.
- Line 31: Please, use italic letters in “Coix lacryma-jobi”.
- Line 41: “…where the cultivation areas reached 50,000 hm2 in 2021”, this paragraph has not been properly justified with a reference.
- Line 42: “The contamination of coix seeds mainly comes from pesticide residues, heavy metal pollution during the production”. It is possible others sources of contaminations? For instance, by mycotoxins?
- Line 51: “And the hazardous nature of toxic pesticide residues…” Please revise this sentence.
6. Line 60: “Moreover, the quality and safety of coix seed were evaluated”. This sentence is inaccurate; in fact quality parameters were not studied. Please correct this sentence to make more precise.
7. Lines 65-67: authors should provide more information about the analysed samples: commercially available, bulk samples, etc.
8. Figure 1: Most of the sample sites were located in the southern of Guizhou province. Why did the authors not take samples in more areas to monitor the entire province?
9. Line 84: in materials and methods section, the instrumentation used is lacking.
10. Lines 84-102: The authors would explain how many replicates of each coix sample were done and insert standard deviation of results.
11. Lines 94, 154: Please, change “fifilter” by “filter”.
12. Line 96: Please, correct “whlie”.
13. Line 104: The authors should rewrite this paragraph “Pollution index” explaining clearly the pollution index studied “Nemerow integrated pollution index (PN)”.
14. Line 152: Use “mL” instead of “ml”.
15. Line 162: I believe that it more correct to use the abbreviation “min” instead of “minute” and should be used consistently throughout the text.
16. Line 168: Please, check the temperature units through the text, thermal values must be separated from the corresponding units: “40 ºC” not “40ºC”.
17. Line 195: ADI is the “acceptable daily intake” or “allowable daily intake”, please use only one.
18. Line 222: Please, change “formula” by “equation”, and should be used consistently throughout all the text.
19. Line 236: Why the authors calculate the “correlation coefficient” instead of “regression coefficient” (R2)?
20. Line 236: Please, insert the reference or manual to check the validation parameters applied and the criteria to evaluate the results obtained.
21. Line 237: Please, revise the caption of the Table 1.
22. Line 271: Please, provide an alternative caption of the Table 3. Like in (CP) provide the meaning of the acronym (CSGB) in the Table 3.
23. Lines 290-291: Please, explain better why the contents of Cr were not evaluated in the process.
24. Line 307: Data of the single pollution index (Pi) was showed in Figure 2 and repeated en Table 4. In Table 4 could be deleted data of Pi
25. Line 348: It is necessary to know the characteristics of the coix samples to evaluate the residual pesticide concentrations in a real context.
26. Line 352: Change “Table 9” by “Table 8”.
27. Figure 4. Data showed in the Figure 4 should be included in the Table 7.
Author Response
尊敬的审稿人:
请参阅附件!

Reviewer 2 Report
The manuscript is of great interest and reflects interesting and careful research. However, some aspects should be taken into account.
1) Introduction: it is too short. The authors should increase the information and its depth. More information on the toxic effects of the metals and pesticides studied is needed. The effects on health and crops and the rate of uptake and accumulation.
2) Material&Methods: the quality control of the method needs to be included. It is not part of the results, but part of the material and methods. Change it.
No statistical analysis has been done? It should appear in the Material and Methods section, including the tests applied and why.
3) Conclusions: they are very brief. What are the future perspectives of this study? Impact?
4) References are scarce. Look for more studies that may be of interest.
Author Response
Dear Reviewers:
Please see the attachment!

Reviewer 3 Report
· _ *In the introduction, I missed some data on the consumption rate of coix seeds in China and on a global scale.
· _*The authors should consider including a research gap in this area at the end of the manuscript.
· _* In lines 395-397: the authors made some valid recommendations. However, this should be written more elaborately. The authors should briefly offer insights on how this can be accomplished. This will offer the manuscript more international relevance.
Author Response

(The authors gave the same response as above.)

Round 2
Reviewer 1 Report
1. There are numerous typographycal errors in the text, for instance: lines 102, 110, 111, 225, 243, 291, 292, 350, 352, 433.
2. Line 272: this reference "(Duncan, 1955)"must be cited according to journal's instructions.
3. In Table 6, at "Risk score" column, data should be have the same decimal numbers, so "21,33" could be changed by "21".
Author Response
亲爱的审稿人:
查看附件!
